# Green Fabrication of Silver Nanoparticles, Statistical Process Optimization, Characterization, and Molecular Docking Analysis of Their Antimicrobial Activities onto Cotton Fabrics

**DOI:** 10.3390/jfb15120354

**Published:** 2024-11-21

**Authors:** Nada S. Shweqa, Noura El-Ahmady El-Naggar, Hala M. Abdelmigid, Amal A. Alyamani, Naglaa Elshafey, Hadeel El-Shall, Yasmin M. Heikal, Hoda M. Soliman

**Affiliations:** 1Botany Department, Faculty of Science, Mansoura University, Mansoura 35516, Egypt; nadasalah2000@mans.edu.eg (N.S.S.); hudasoliman@mans.edu.eg (H.M.S.); 2Department of Bioprocess Development, Genetic Engineering and Biotechnology Research Institute (GEBRI), City of Scientific Research and Technological Applications (SRTA-City), New Borg El Arab City 21934, Egypt; nelahmady@srtacity.sci.eg; 3Department of Biotechnology, College of Science, Taif University, Taif 21944, Saudi Arabia; h.majed@tu.edu.sa (H.M.A.); a.yamani@tu.edu.sa (A.A.A.); 4Botany and Microbiology Department, Faculty of Science, Arish University, Al-Arish 45511, Egypt; n_fathi@aru.edu.eg; 5Environmental Biotechnology Department, Genetic Engineering and Biotechnology Research Institute (GEBRI), City of Scientific Research and Technological Applications (SRTA-City), New Borg El Arab City 21934, Egypt; hadeel.elshall28@gmail.com

**Keywords:** nanoparticles, silver, green synthesis, *Aspergillus flavus*, statistical optimization, characterization, cotton fabrics, antimicrobial activity, molecular docking analysis

## Abstract

Nanotechnological methods for creating multifunctional fabrics are attracting global interest. The incorporation of nanoparticles in the field of textiles enables the creation of multifunctional textiles exhibiting UV irradiation protection, antimicrobial properties, self-cleaning properties and photocatalytic. Nanomaterials-loaded textiles have many innovative applications in pharmaceuticals, sports, military the textile industry etc. This study details the biosynthesis and characterization of silver nanoparticles (AgNPs) using the aqueous mycelial-free filtrate of *Aspergillus flavus*. The formation of AgNPs was indicated by a brown color in the extracellular filtrate and confirmed by UV-Vis spectroscopy with a peak at 426 nm. The Box-Behnken design (BBD) is used to optimize the physicochemical parameters affecting AgNPs biosynthesis. The desirability function was employed to theoretically predict the optimal conditions for the biosynthesis of AgNPs, which were subsequently experimentally validated. Through the desirability function, the optimal conditions for the maximum predicted value for the biosynthesized AgNPs (235.72 µg/mL) have been identified as follows: incubation time (58.12 h), initial pH (7.99), AgNO_3_ concentration (4.84 mM/mL), and temperature (34.84 °C). Under these conditions, the highest experimental value of AgNPs biosynthesis was 247.53 µg/mL. Model validation confirmed the great accuracy of the model predictions. Scanning electron microscopy (SEM) revealed spherical AgNPs measuring 8.93–19.11 nm, which was confirmed by transmission electron microscopy (TEM). Zeta potential analysis indicated a positive surface charge (+1.69 mV), implying good stability. X-ray diffraction (XRD) confirmed the crystalline nature, while energy-dispersive X-ray spectroscopy (EDX) verified elemental silver (49.61%). FTIR findings indicate the presence of phenols, proteins, alkanes, alkenes, aliphatic and aromatic amines, and alkyl groups which play significant roles in the reduction, capping, and stabilization of AgNPs. Cotton fabrics embedded with AgNPs biosynthesized using the aqueous mycelial-free filtrate of *Aspergillus flavus* showed strong antimicrobial activity. The disc diffusion method revealed inhibition zones of 15, 12, and 17 mm against *E. coli* (Gram-negative), *S. aureus* (Gram-positive), and *C. albicans* (yeast), respectively. These fabrics have potential applications in protective clothing, packaging, and medical care. In silico modeling suggested that the predicted compound derived from AgNPs on cotton fabric could inhibit Penicillin-binding proteins (PBPs) and Lanosterol 14-alpha-demethylase (L-14α-DM), with binding energies of −4.7 and −5.2 Kcal/mol, respectively. Pharmacokinetic analysis and sensitizer prediction indicated that this compound merits further investigation.

## 1. Introduction

Nanotechnology, a rapidly expanding field, is focused on the fabrication and engineering of materials at an incredibly small scale of less than one micron [1]. The nanoparticles exhibit distinct properties from the bulk material because of their tiny size, quantum related effects, and high surface area-to-volume ratio [2]. Among them, silver nanoparticles (AgNPs) have attracted increasing interest due to their unique physicochemical characteristics. These characteristics include exceptional thermal and electrical conductivity, longer chemical stability, unique catalytic properties, their interaction with light through surface plasmon resonance, and unique optical properties [3]. The global industrial production of AgNPs has significantly increased, with predictions suggesting that the annual production will reach 800 tons by 2025 [4].

AgNPs have a wide range of potential applications, including agriculture, medicine, engineering, and environmental sciences. AgNPs applications include biosensors, bio-labeling materials, surface coating and medical device coating, optical receptors, catalysts in chemical and biochemical reactions, electronics, cosmetics, textiles [5,6]. AgNPs properties include antibacterial, antifungal, antiviral, anti-inflammatory, anti-angiogenic, and antitumor activities [5].

Traditionally, AgNPs have been fabricated employing chemical reduction and physical methods. The production of nanoparticles via traditional methods is extremely costly, and generates toxic, hazardous chemicals that pose environmental risks. Moreover, the potential sedimentation of chemicals on the surfaces of nanoparticles produced via these methods reduces their predicted purity. Consequently, synthesizing silver nanoparticles of the desired size may pose challenges, requiring an additional step to avoid particle aggregation [7]. Compared to conventional methods, green synthesis of nanoparticles offers a more feasible, non-toxic, eco-friendly and cost-effective alternative. For the green synthesis of nanoparticles, plant extract, fungi, yeast, bacteria, and even plant-derived materials were used as a reducing, capping, and stabilizing agent to aid in the green synthesis of AgNPs [8]. It has the potential to produce nanoparticles with diverse characteristics, which could lead to new and interesting applications [9]. Fungi are exceptional producers of diverse metabolites including amino acids, proteins, and enzymes. These substances facilitate the environmentally friendly production of nanoparticles (NPs) and enhance their stability. Furthermore, fungi present several benefits: they are easily managed, scalable for production, yield substantial biomass, exhibit minimal toxicity, and can generate NPs both internally and externally [10]. Fungal strains were used to synthesize eco-friendly metal and metal oxide nanoparticles including Ag, Au, Pt, Si, ZnO, ZrO_2_, Zr, TiO_2_, and Ti [11,12]. Fouda, et al. [13] highlighted the great efficacy of *Aspergillus flavus* as a biocatalyst for the green synthesis of AgNPs.

The factor-by-factor technique, commonly referred as the traditional technique, is the first method for optimizing the biosynthesis process of nanoparticles. It involves changing one independent variable while keeping the other variables at their optimal values in order to identify the appropriate conditions [14]. The traditional factor-by-factor method has several disadvantages, including being time-consuming, expensive, labor-intensive, and requiring a lot of reagents and materials. Additionally, the interaction effects between the independent variables are ignored by the factor-by-factor technique [15]. A variety of mathematical and statistical methods can be applied to optimize multiple variables simultaneously. In contrast to conventional methods, Box-Behnken design (BBD) is a highly efficient response surface mathematical design that facilitates the elucidation of relationships between the studied variables and responses, optimizing multiple variables simultaneously, minimizing the overall experimental trials, defining the optimal process conditions, and maintaining a high degree of accuracy in the final result. Moreover, it is cost-effective, highly adaptable, and faster [16].

Cotton textiles are known for their excellent air permeability, high moisture absorption, and pleasant tactile sensations. Nanotechnology can enhance cotton garments with advanced features [17,18]. Zhou, et al. [19] demonstrated that plasma treatment can generate AgNPs on cotton fabric, improving its antibacterial properties. Additionally, combining silver nitrate (AgNO_3_) and *Aloe vera* extract can imbue cotton fabrics with ultraviolet protection and antimicrobial activity through the in-situ production of AgNPs. In situ synthesis of AgNPs achieves appropriate particle deposition by eliminating the need for a separate reaction period [20].

Cellulose has shown no antibacterial properties against Gram-positive or Gram-negative bacteria, which has restricted its use in antibacterial applications. However, this polysaccharide can be modified by incorporating antibacterial agents, such as nanomaterials, to enable its use in the pharmaceutical, food packaging, and cosmetic industries [21]. In addition, cellulose can be modified by combining it with other polymers, nanoparticles, or biomaterials to produce composite materials. These composites exhibit enhanced characteristics such as greater tensile strength, superior barrier properties, and increased biocompatibility, making them suitable for applications in environmental cleanup, medical devices, and packaging [22].

Nonetheless, the high water content in cotton enriched with cellulose makes fibers more sensitive to microbial attack and creates an ideal environment for bacterial and fungal growth [23,24]. AgNPs, which have long been recognized as effective biocides with antibacterial activity, now have more applications in textiles. Smaller particle sizes result in greater antimicrobial effectiveness [25]. Furthermore, the effectiveness of AgNPs in combating microbial infections is attributed to their positive charge, which enables them to interface and damage the negatively charged microbial plasma membrane [26]. This interaction effectively inhibits microbial growth and prevents the spread of infection. In the realm of medical devices, coatings containing AgNPs can improve their performance by endowing them with antimicrobial properties. Jain, et al. [27] demonstrated that fabrics treated with AgNPs exhibited the strongest antibacterial effect against *Bacillus licheniformis*, with 93.3% inhibition. The same fabric displayed moderate effectiveness against *Klebsiella pneumoniae* and *Escherichia coli* with inhibition rates of 20% and 10%, respectively. Patil, et al. [28] found that cotton fabrics infused with AgNPs have outstanding antibacterial properties against gram-positive and gram-negative microorganisms, such as *Bacillus subtilis, Pseudomonas aeruginosa*, *Staphylococcus aureus*, and *Escherichia coli*.

Silver nanoparticles (AgNPs) have been confirmed as an amazing antimicrobial agent due to their broad-spectrum antimicrobial actions against Gram-positive and Gram-negative bacteria, fungi, viruses, and mycobacteria. When microorganisms are exposed to AgNPs, silver ions are continuously released from the nanoparticles. Microorganisms are killed by the constant release of silver ions from AgNPs, free silver ions absorbed by cells inhibit respiratory enzymes, which results in the production of reactive oxygen species (ROS) that interfere with the synthesis of adenosine triphosphate. ROS serves as the primary agents responsible for inducing DNA alterations and damaging cell membranes [29]. Also, silver ions readily adhere to the cell wall and cytoplasmic membrane due to their close association with sulfur proteins and the influence of electrostatic attraction [30]. The binding of AgNPs to microbial membranes induces irreversible morphological alterations in the cell membrane structure and the changes in cell structure can lead to increased permeability of the cell membrane, subsequently impacting the cell’s ability to regulate its activities effectively [31]. Molecular docking is utilized to determine ligand and receptor interactions. Although prediction tools for computational synthesis are currently not widely integrated, this situation is anticipated to evolve rapidly as the field progresses rapidly [32]. Quantitative Structure-Activity Relationship (QSAR) modeling serves as a vital computational method in medicinal chemistry and toxicology research. This method is instrumental for identifying new bioactive compounds and evaluating their chemical safety. The models generated through QSAR were then applied to forecast the desired properties of unexplored or untested substances [33].

Our study focused on the green synthesis of AgNPs using the aqueous mycelial-free filtrate of *Aspergillus flavus*. To optimize the process, BBD was employed, considering four crucial factors: incubation duration (X_1_ in hours), starting pH (X_2_), silver nitrate concentration (X_3_ in mM/mL), and temperature (X_4_ in °C). The biosynthesized AgNPs were then characterized, and their antibacterial efficacy was assessed using the disc diffusion technique with AgNPs-loaded cotton fabrics. Additionally, computational and forward synthesis techniques was utilized to anticipate the interaction between AgNPs produced by the aqueous mycelial-free filtrate of *Aspergillus flavus* and cotton fabrics, regarded as the most unadulterated form of cellulose, containing roughly 90% cellulose. The study also aimed to predict the resulting compound’s potential for human skin sensitization and pharmacokinetic characteristics.

## 2. Materials and Methods

### 2.1. Fungal Isolate and Biomass Preparation

*Aspergillus* sp. obtained from the Mycology Laboratory at Mansoura University’s Faculty of Science, Egypt, was cultured on Potato Dextrose Agar (PDA) plates prepared by combining potato extract (200 g), dextrose (20 g), and agar (20 g) in 1 L of distilled water, followed by autoclaving at 121 °C for 20 min. The cultured *Aspergillus* sp. was incubated at 25 °C for five days and stored at 4 °C for subsequent AgNPs synthesis.

### 2.2. Fungal Morphological and Molecular Identification

Two microscopic methodologies were used to determine the fungus’s morphological characteristics: (1) The fungal isolate’s vegetative and reproductive structures were examined using a digital light microscope (Optika Model B-380 Ver. 2, Italy) with 100× and 400× magnification and (2) The isolate was examined using scanning electron microscopy (SEM) (JSM-6510 L.V, JEOL Ltd., Tokyo, Japan) at the Electron Microscope Unit at Mansoura University in Egypt. After drying, the specimen was mounted on metallic stubs and sputter-coated with a thin layer of gold prior to imaging [34]. *Aspergillus* sp. was then identified using 18S rDNA gene sequence analysis. The extraction of DNA from the fungal isolate and the amplification of the 18S rDNA gene region by PCR are the steps in this method. Subsequently, the precise fungal species was identified by comparing the DNA sequence obtained to the NCBI GenBank database.

DNA was isolated using the CTAB technique, as outlined by Doyle [35] to obtain DNA from the fungal mycelium. The Internal Transcribed Spacer (ITS) region was amplified for molecular identification of *Aspergillus* sp. using Primer3 software (https://primer3.org/ (accessed on 1 August 2023)) [36] designed primers 5′-AGAAACGGCTACCACATCCA-3′ (forward) and 5′-TCTGGACCTGGTGAGTTTCC-3′ (reverse), synthesized by BIONEER Inc. (Oakland, CA, USA) targeting this conserved fungal identification region. The 18S rDNA region of the selected isolate was PCR-amplified in a 50 μL reaction containing 25–50 ng DNA, 50 mM primers, 0.5 U/μL Taq DNA polymerase, and 0.2 mM each dNTP. The protocol included denaturation at 94 °C for 6 min; 35 cycles of denaturation at 94 °C for 45 s, annealing at 56 °C for 45 s, and synthesis at 72 °C for 45 s; and a final extension at 72 °C for 5 min. PCR products were stored at −20 °C and visualized on a 2% agarose gel with 0.5 μg/mL ethidium bromide using a MicroDoc gel documentation system (Cleaver Scientifc Ltd., Rugby, UK).The 18S rDNA gene sequence, obtained using an ABI 3730XL DNA Analyzer (Applied Biosystems, Waltham, MA, USA) and edited with Finch (version 1.4.0), was analyzed using BLAST (http://blast.ncbi.nlm.nih.gov (accessed on 30 September 2023)). Mega 11 [37] was used to construct phylogenetic tree. A comparison of the fungus’s 18S rDNA gene sequence to known sequences in the NCBI GenBank database was conducted using BLASTn and neighbor-joining to investigate its evolutionary relationships [38], The sequenced gene was deposited in the database for future reference and sharing.

### 2.3. Extracellular Synthesis of AgNPs Using A. flavus

Extracellular AgNPs synthesis using *A. flavus* involves cultivating fungal biomass on PDA plates. After incubation, the biomass was filtered and washed with sterile dH_2_O to eliminate the residual media, and 10 g (wet weight) of this biomass was incubated in 100 mL of sterile deionized H_2_O for 24 h at 25 ± 1 °C with agitation at 150 rpm. In order to prevent the photooxidation of silver ions, AgNO_3_ was added in the dark to achieve a final concentration of 1 mM. Control flasks with the aqueous mycelial-free filtrate of *A. flavus* were prepared under the same conditions but without Ag^+^ ions. Different AgNPs concentration series (10–100 μg/mL) was obtained by diluting AgNPs stock solution (500 μg/mL) with deionized water. Those standards were measured together with samples to obtain the corresponding UV-Vis and the calibration curve for calculating sample concentrations.

### 2.4. The Optimization of AgNPs Biosynthesis: A Box-Behnken Design Approach

To optimize the biosynthesis of AgNPs using the aqueous mycelial-free filtrate of *A. flavus*, the Box-Behnken design was employed. The goal of this method is to investigate the interactions between the process variables and optimize the response (amount of AgNPs yield). Four independent variables were assessed: incubation time (h) (X_1_), initial pH (X_2_), AgNO_3_ concentration in mM/mL (X_3_), and temperature (°C) (X_4_). Each variable was tested at three levels (−1, 0, and 1). In total, 29 experimental runs were carried out. To analyze the results, a second-order polynomial equation was utilized. This equation mathematically delineates the connection between the coded independent variables and anticipated AgNPs yield:

The following second-order polynomial equation was applied to analyze the relationship between the independent variables and the predicted AgNPs:(1)$$Y=\beta_{0}+\sum_{i}\beta_{i}X_{i}+\sum_{ii}\beta_{ii}X_{i}^{2}+\sum_{ij}\beta_{ij}X_{i}X_{j}$$
Y: Predicted AgNPs yield; $\beta_{0}$: Regression coefficient; $\beta_{i}$: Linear coefficients; $X_{i}\;and\;X_{j}$: Levels of the independent variables; $\beta_{ii}$: Quadratic coefficients; $\beta_{ij}$: Interaction coefficients.

### 2.5. Characterization of Bio-Synthesized AgNPs

Various techniques have been employed to characterize the biosynthesized AgNPs using the aqueous mycelial-free filtrate of *A. flavus*. These techniques offer crucial insights into the shape, dimensions, surface chemistry, crystallinity, and stability of the NPs.

#### 2.5.1. Ultraviolet-Visible (UV-Vis) Spectral Analysis

AgNPs formation was assessed using UV-Vis spectroscopy following the method described by Basavaraja, et al. [39]. A Uni Camry UV-VIS Spectrometer UV2 (USA) was used to measure the absorption spectrum of the AgNPs solution between 300 and 700 nm, with deionized water as a blank for background correction. A characteristic peak in the visible spectrum, termed surface plasmon resonance (SPR), indicates the formation of AgNPs.

#### 2.5.2. Scanning Electron Microscopy (SEM) Analysis

SEM was used to analyze the dimensions and structure of AgNPs. The sample was prepared by depositing a drop of the centrifuged AgNPs suspension onto a clean glass surface and allowing water to evaporate, leaving AgNPs on the substrate. SEM analysis, performed with a JEOL JSM-6510 L.V low-vacuum SEM at an accelerating voltage of 30 kV, revealed details regarding the size, shape, and surface characteristics of individual AgNPs.

#### 2.5.3. Transmission Electron Microscopy (TEM) Analysis

TEM provides high-resolution images of AgNPs, enabling detailed analysis of their size and structure. A drop of the AgNPs suspension was placed on a carbon-coated grid (Type G 200, 3.05 μm diameter, EMS, Hatfield, PA, USA) and dried under an infrared lamp. The sample was then examined with a JEOL JEM 2100 (Tokyo, Japan) microscope at Mansoura University, Egypt, operating at 200 kV. This method, as described by Wang [40], offers vital insights into the size distribution, morphology, and crystalline structure of AgNPs.

#### 2.5.4. Selected Area Electron Diffraction (SAED) Pattern

SAED analysis was employed to examine the crystalline structure of the produced AgNPs. The SAED pattern shows concentric diffraction rings, which describe the crystal planes and lattice structure of the NPs [38]. This technique usually requires a very thin sample with a thickness of approximately 100 nm and high-energy electrons with an energy of 100–400 keV to interact with the material as waves instead of particles.

#### 2.5.5. Zeta Potential Distribution

The Zeta potential of the AgNPs was detected utilizing a Zeta Potential Analyzer (Zetasizer ver. 7.01; Malvern Instruments, Westborough, MA, USA) to determine their stability. Specifically, 5 mg of AgNPs were combined with 5 mL of ultrapure water and stirred at room temperature for 30 min. Zeta potential was measured using an EM unit at Mansoura University, Egypt, at a 90° detection angle. This value reflects the surface charge of the NPs, and a high absolute value (+ve or −ve) indicates good colloidal stability owing to electrostatic repulsion between particles.

#### 2.5.6. Energy-Dispersive X-Ray Spectroscopy (EDX) Analysis

Energy-dispersive X-ray Spectroscopy (EDX) was used to confirm the composition of the AgNPs using an X-ray microanalyzer (Oxford 6587 INCA) attached to a JEOL JSM-5500 LV SEM at 20 kV. The EDX spectrum, taken in the spot profile mode from a densely populated AgNPs area on the sample surface [41], showed a significant Ag signal, verifying the presence of metallic Ag in the NPs.

#### 2.5.7. X-Ray Diffractometer (XRD) Analysis

XRD analysis was employed to detect the crystalline structures of the produced dried powder of AgNPs. The analysis was performed with XRD (Shimadzu Xlab 6100, Japan) at Japan University of Science and Technology, Egypt. The values of 2θ were measured between 20° to 80°. The X-ray diffractometer was operated with continuous scanning at 40 kV, a current of 30 mA and a speed of scanning of 12°/min.

#### 2.5.8. Fourier Transform Infrared (FTIR) Spectroscopy

FTIR spectroscopy, performed at a 4 cm^−1^ resolution over wavenumbers from 400 to 4000 cm^−1^, identified functional groups on the biosynthesized AgNPs, revealing adsorbed biomolecules that influence their stability and biological activity. The samples were prepared by dispersing AgNPs in a dry KBr matrix, compressing them into a clear disc, and analyzing them using a JASCO FTIR spectrometer (Tokoyo, Japan) with a KBr pellet standard at the Faculty of Science, Mansoura University, Egypt.

### 2.6. Evaluation of Antimicrobial Activity Using AgNPs-Loaded Cotton Fabrics

Cotton fabrics were employed as an innovative delivery mechanism for the biosynthesized AgNPs to assess their antimicrobial effectiveness. The cotton fabrics were subjected to comprehensive washing, disinfection, and air-drying procedures, following the methods outlined by Durán, et al. [42]. Subsequently, AgNPs were applied to the fabric surface and then cut into 1 cm × 1 cm squares [43]. The fabric pieces were left to dry in the air. The effectiveness of AgNPs -loaded fabrics was tested against Gram-positive bacteria (*Staphylococcus aureus*), Gram-negative bacteria (*Escherichia coli*), and yeast (*Candida albicans*) cultured on Luria-Bertani (LB) agar plates. The disc diffusion technique was applied to assess the antimicrobial potency. AgNPs-loaded fabric squares measuring 1 cm × 1 cm were positioned on the inoculated agar plates. To determine the influence of AgNPs independently, a control group (sterile cotton fabrics without AgNPs) were used. The inoculated plates were incubated at 30 °C. Following the incubation time, the plates were assessed for the presence of inhibition zones (clear areas surrounding the fabric squares). The inhibition zones demonstrate the suppression of microbial growth by AgNPs. The diameter of these zones was measured in millimeters to quantify the efficacy of antibacterial activity [43].

### 2.7. Molecular Docking Approach

#### 2.7.1. Ligand Preparation

From the PubChem database, the 3D structure of cellulose (Pumchem CID: 16211032), which has the chemical formula C1_2_H_22_O_11_, was retrieved in the SDF format. Following this, Avogadro 1.2.0 software [44] was employed to conduct energy minimization using the MMFF94 force field.

#### 2.7.2. Forward Reaction Prediction

To gain insight into the various combined reactions between cellulose (in the form of cotton fabric) and biosynthesized AgNPs, a forward reaction prediction method was employed. The ASKCOS MIT [45] system (https://askcos.mit.edu/forward?tab=forward (accessed on 30 September 2024)) was used to forecast all the potential forward reactions. The process utilized all previously prepared files in SDF format for the reaction, with default settings.

#### 2.7.3. Protein Preparation

Penicillin-binding proteins (PBPs) are key targets of beta-lactam antibiotics and have an essential role in the synthesis and maintenance of bacterial cell walls. These proteins are located in the periplasm and are associated with the membranes. Bacteria have developed various resistance strategies, including modification of target enzymes to decrease their attraction to beta-lactam antibiotics. We examined a protein structure from the database, identified as PDB ID: 1QMF, with a Resolution of 2.80 Å, which was crystallized using X-ray diffraction techniques [46]. In addition, we investigated lanosterol 14-alpha-demethylase (L-14α-DM), an enzyme that functions as a valuable pharmacological target for a wide spectrum of anticandidal and antifungal treatments in eukaryotic organisms, including humans. This enzyme is effective against commensal and pathogenic fungi. For this part of the study, we analyzed the protein structure from the database labeled PDB ID: 5V5Z, with a resolution of 2.90 Å, also crystallized through X-ray diffraction methods [47].

#### 2.7.4. Molecular Docking Process

AutoDockTools-1.5.7 Świątek, et al. [48] was utilized to simulate the attachment of biologically synthesized AgNPs to cotton fabrics at the protein-binding site. The results of the ligand-target docking process were subsequently cataloged based on the binding energy (∆G) in kcal/mol. BIOVIA Discovery Studio 2021 [46] was employed to depict the molecular docking interactions between the protein receptors and docked predicted molecules.

#### 2.7.5. ADMET, Pharmacokinetics, and Skin Sensitization Prediction of Products Generated from AgNPs-Loaded Cotton Fabrics

The online computational platform Swiss-ADME (http://www.swissadme.ch/ (accessed on 1 October 2024)) was used to determine ADMET and pharmacokinetic profiles [49]. Beginning with the SMILES formula, this tool enables the calculation of a chemical compound’s physicochemical characteristics, pharmacokinetic properties, and drug-likeness with a predictive accuracy between 72% and 94%. Furthermore, the Pred-Skin computational tool was used to evaluate chemically induced skin sensitization [50].

### 2.8. Statistical Analysis

The optimization design was generated using Design-Expert software (Version 7.0.0, created by Stat-Ease, Inc., Minneapolis, MN, USA). Experimental data were analyzed using ANOVA via multiple regression analysis, determining *p*-value, *f*-value, and confidence levels, alongside the calculation of the determination coefficient (*R*^2^) and adjusted *R*^2^. STATISTICA software (Version 8, StatSoft, Inc., Tulsa, OK, USA) was utilized to generate three-dimensional surface plots for visualization purposes.

## 3. Results and Discussion

Nanoparticles, which typically range in size from 1 to 100 nm, are an interesting class of materials in nanotechnology. Although traditional chemical and physical synthesis approaches usually require tight control measures, this study investigates a potential alternative: a biological approach to nanoparticles synthesis. This alternative approach has various advantages over traditional methods. First, biological processes have the ability to produce NPs with an uniform size distribution, which is critical for optimal performance in various applications [51]. Second, biological methods provide a significant cost advantage over chemical and physical procedures for NPs synthesis. Third, these methods frequently reduce the usage of harsh chemicals, lessening the environmental impact [51]. Finally, biologically synthesized nanoparticles may be more biocompatible than chemically generated NPs.

Fungi are emerging as powerful bio-synthesizers for NPs due to their ability to produce enormous quantities of extracellular enzymes that might facilitate downstream processing and improve nanoparticles characteristics. In comparison to bacteria, fungi are capable of producing a greater quantity of NPs due to their increased production of proteins, which are involved in the synthesis of nanoparticles. Consequently, fungi constitute a sustainable and promising source for nanoparticles production [52]. AgNPs were synthesized in this study using the aqueous mycelial-free filtrate of *A. flavus* in a simple, rapid, environmentally friendly, and cost-effective approach. The metabolites reduced and stabilized Ag^+^ ions, transforming them into Ag^0^. Furthermore, the production of Ag-NPs using *A. flavus* was previously reported by Gopa and Pullapukuri [53].

### 3.1. Identification of A. flavus

*Aspergillus* sp. strain SF-4 was first identified according to the macroscopic and microscopic morphological features. Molecular identification was performed to identify isolate at the species level. *Aspergillus* sp. strain SF-4 displayed a yellowish green aerial hyphal growth (Figure 1A). The microscopic characteristics of the fungus are depicted in Figure 1B,C displaying septate, branched mycelium, vesicle, and conidia. Figure 1D shows the SEM image of *Aspergillus* sp. strain SF-4, produced abundant conidiophores with abundant spherical conidia in long chains.

At the molecular level, the identification of fungal species is shown in Figure 2. To accomplish this, genomic DNA was extracted from the mycelia of the fungus and the ITS gene region was amplified using specific primers. The resulting PCR products were then purified and sequenced. Subsequently, the BLAST search tool was employed for comparison of the obtained sequence data with all the available fungal sequences from the GenBank database. The phylogenetic analysis in Figure 2 indicated that the present sequences are identical to the ITS sequences of *Aspergillus flavus* strain SF-4. The phylogenetic tree has been divided into two primary groups. One of these groups was further divided into two subgroups, one of which contained our isolate with accession number MH511139 and five additional species. The isolate was most closely related to the *Aspergillus flavus* strain I.P-7 (MH511107) species with 98% matching, and the second cluster consisted of three species. According to the study of ITS gene sequence, together with its phenotypic characteristics, the *Aspergillus* sp. strain SF-4 (MH511139) was identified as *Aspergillus flavus* strain SF-4. These results were agreed with Derbalah, et al. [54].

### 3.2. Extracellular Synthesis of AgNPs Using A. flavus

Figure 3A,B show that the reaction mixture turned brown, indicating the biosynthesis of AgNPs [44].The intensity of the color correlates with the number of electrons generated from the reduction of NO_3_ to NO_2_, facilitating the conversion of Ag^+^ into metallic ions (Ag^0^) [45]. UV/VIS spectral analysis was conducted to characterize the biosynthesized AgNPs, covering the range 300–700 nm, as depicted in Figure 3C. The AgNPs displayed a characteristic single peak in their optical absorption spectrum, with the maximum absorbance occurring at 426 nm. Brause, et al. [55] proposed that the optical absorption spectrum of metallic nanoparticles is primarily influenced by the surface plasmon resonance (SPR) phenomenon. They also observed a direct correlation between the absorption peak location and the particles dimensions. Research by Fouda, et al. [13] demonstrated that while the absorption increased, the maximum SPR peak remained constant at 415 nm. They explained this phenomenon as a result of the total elimination of Ag^+^ ions, which led to an intensification of color after 15 days.

### 3.3. Optimizing AgNPs Biosynthesis Using Box-Behnken Design (BBD)

In the current study, the effects of four bioprocess variables on the biosynthesis of AgNPs using the aqueous mycelial-free filtrate of *A. flavus* were assessed. These variables were the AgNO_3_ conc. (mM), initial pH level, temperature (°C), and incubation time (h). BBD was used to determine the optimal values for these variables that maximize the biosynthesis of AgNPs. As well as, to investigate the individual, interaction, and quadratic effects of these variables on the biosynthesis of AgNPs. Table 1 shows that each variable was assessed at three distinct levels (−1, 0, and 1). BBD involved 29 experimental runs, with six of these (runs 1, 2, 10, 13, 27, and 28) situated at the central point. According to Table 1, run No. 25 yielded the lowest AgNPs biosynthesis of 8.19 µg/mL, which was achieved under the following conditions: pH of 7, incubation time of 48 h, silver nitrate concentration of 1 mM/mL, and a temperature of 25 °C. Conversely, the highest biosynthesis of AgNPs achieved was 235.03 µg/mL in run No. 22, which was performed under the following conditions: pH of 8, an incubation duration of 72 h, a silver nitrate concentration of 3 mM/mL, and a temperature of 30 °C. Table 1 shows the results obtained for both the synthesized AgNPs and the anticipated values. Notably, the actual results closely matched the predicted outcomes for the AgNPs.

### 3.4. Model Fitting

The BBD data for the biosynthesized AgNPs using the aqueous mycelial-free filtrate of *A. flavus* was subjected to statistical analysis through multiple regression and analysis of variance (ANOVA). The ANOVA results are summarized in Table 2. The coefficient of determination (*R*²) serves as a statistical metric to assess the model’s fit. It quantifies the proportion of variance in the values of the response variable that can be attributed to the independent experimental variables and their mutual interactions [56]. According to the estimated *R*^2^ value of 0.9994, the independent selected variables were responsible for 99.94% of the variance in the biosynthesized AgNPs, which were synthesized using the aqueous mycelial-free filtrate of *A. flavus*. Consequently, the model is unable to effectively describe only 0.06% of the total variance in the biosynthesized AgNPs. The *R*^2^ value for a regression model generally falls between 0 and 1. A higher *R*^2^ value, closer to 1, indicates that the model is more effective in predicting the response yield [57]. A regression models having an *R*^2^ value greater than 0.9 was considered to be highly correlated [58]. The value of the adjusted *R*^2^ was 0.9988, while the predicted *R*^2^ was 0.9976. To ascertain the model’s significance and precision, the predicted *R*^2^ and adjusted *R*^2^ values must be within 20% of each other, indicating a substantial degree of agreement between the two metrics [59].

Table 2 represents the significance of each coefficient and the interactions effects between various factors, determined probability values (*p*-values) and *f*-values. Lower *p*-values indicate higher significance of the corresponding coefficient. In addition, our findings indicated that the process variables with *p*-values < 0.05 significantly influencing the biosynthesis of AgNPs. Process variables with *p*-values less than 0.05 were considered to exert a significant effect on the response, hence validating the validity of the model terms [60].

The model *f*-value of 1734.16 and a *p*-value < 0.0001, suggesting a high level of significance. The *p*-values less than 0.05 suggest that the linear coefficients of AgNO_3_ conc. (X_1_), initial pH level (X_2_), temperature (X_3_), and incubation time (X_4_) are statistically significant and may function as limiting variables for the rate of biosynthesis of AgNPs with slight variations in their values. The *f*-values for X_1_, X_2_, X_3_, and X_4_ were 39.79, 11,471.09, 3146.50, and 1139.43; respectively. The quadratic coefficients (X_1_^2^, X_2_^2^, X_3_^2^ and X_4_^2^) had significant *p*-values of <0.0001 and *f*-values of 971.13, 4741.96, 83.53, and 72.15, respectively. Based on the *p*-values of the coefficient, it can be concluded that the interaction effects among combinations of X_1_ X_2_, X_1_ X_4_, X_2_ X_3_, X_2_ X_4_ and X_3_ X_4_ are significant (*p* < 0.0001). Nevertheless, the interaction effect between AgNO_3_ concentration and temperature is not statistically significant.

Moreover, the signs (+ or −) of the calculated coefficients were utilized to understand their influence on the response variable yield [61]. The positive coefficients of linear effects for all variables (X_1_, X_2_, X_3_, X_4_) indicated an increase in the AgNPs biosynthesis. Positive coefficients indicate synergistic interactions between two variables. While negative coefficients indicate antagonistic interactions between two variables.

The positive coefficients of the interaction effects between the different combinations of X_1_ X_2_, X_2_ X_3_, X_2_ X_4_, X_3_ X_4_ and the quadratic coefficients (X_2_^2^, and X_4_^2^) suggested an increase in AgNPs biosynthesis. Furthermore, the negative coefficients for the interaction effects between X_1_ X_3_, X_1_ X_4_, as well as the quadratic coefficients (X_1_^2^, X_3_^2^), suggested a reduction in the AgNPs biosynthesis (Table 2). Notably, the statistically significant positive coefficients for the interaction effects between different variables (*p* < 0.0001) demonstrated their significant contribution to the enhancement of AgNPs biosynthesis. The model’s accuracy was assessed using the signal-to-noise ratio, which can be analyzed by checking the adequate precision value. This value must exceed four in order to suggest a robust model. The current model’s adequate precision value is 167.47.

Table 3 presents the fit summary results. The fit summary was employed to identify the most appropriate model for synthesizing AgNPs using the aqueous mycelial-free filtrate of *A. flavus* among linear, two-factor interaction (2FI), and quadratic models. The fit summary results indicate that the quadratic model is the most significant model and the most appropriate model for synthesizing AgNPs using the aqueous mycelial-free filtrate of *A. flavus* with insignificant lack-of-fit test (*f*-value = 0.65, *p*-value = 0.735) and *p*-value < 0.0001. The quadratic model displayed the highest values of *R*^2^ adj. *R*^2^ and predicted *R*^2^ values of 0.9994, 0.9988 and 0.9976; respectively and the lower standard deviation value (1.87).

To explore the relationships between the independent variables [AgNO_3_ concentration (X_1_), initial pH (X_2_), temperature (X_3_), and incubation time (X_4_)] and the dependent variable [predicted AgNPs biosynthesis (Y)], a second-order polynomial equation was employed. The following equation allows for the determination of optimal values for each independent variable that would lead to the highest predicted AgNPs biosynthesis.
Y = 72.88+ 3.41X_1_ + 57.95X_2_ + 30.35X_3_ +18.26X_4_ + 12.32X_1_X_2_ − 1.93X_1_X_3_ − 4.18X_1_X_4_ + 17.61X_2_X_3_ + 27.26X_2_X_4_ + 6.72X_3_X_4_ − 22.93X_1_² + 50.68X_2_² −6.73X_3_² + 6.25X_4_²(2)

#### 3.4.1. Three-Dimensional (3D) Surface Plot

The three-dimensional graphs are tools that are used for explaining the interactions between the various variables and to predict the most favorable conditions for the maximum response [62]. 3D response surface plots were applied in this study to investigate the pairwise interactions between initial pH level, time of incubation, concentration of AgNO_3_, and temperature, to identify the optimal conditions for maximizing AgNPs biosynthesis (Figure 4). El-Naggar, et al. [63] reported that the temperature, initial pH level, silver nitrate concentration, and incubation time are crucial in AgNPs biosynthesis.

The 3D surface graphs (Figure 4A–C) illustrate the effect of AgNO_3_ concentration on AgNPs biosynthesis using the aqueous mycelial-free filtrate of *A. flavus* when interacting with the other three factors: initial pH level, temperature and incubation period; respectively. The 3D surface plots indicate that AgNPs biosynthesis increased by the increasing the AgNO_3_ concentration. The maximum AgNPs biosynthesis was obtained toward the center level of the AgNO_3_ concentration (~3 mM/mL).

The 3D surface graphs (Figure 4A,D,E) illustrate the effect of initial pH level on AgNPs biosynthesis using the aqueous mycelial-free filtrate of *A. flavus* when interacting with the other three factors: AgNO_3_ concentration, temperature, and incubation time; respectively. These plots reveal a positive correlation between initial pH and AgNPs yield. The maximum biosynthesis of AgNPs was found at an initial pH level of approximately 8. Several studies highlighted the significant impact of pH on the size, shape, and stability of AgNPs [64]. In this context, Khan and Jameel [65] reported that the maximum AgNPs was obtained at pH 8. In contrast, Singh, et al. [66] synthesized spherical AgNPs (12–17 nm) from the extract of hibiscus leaves at pH 6.

The 3D surface graphs (Figure 4B,D,F) illustrate the effect of temperature on AgNPs biosynthesis using the aqueous mycelial-free filtrate of *A. flavus* when interacting with the other three factors: AgNO_3_ concentration, initial pH level and incubation time; respectively. The biosynthesis of AgNPs was found to increase as the temperature increased, as evidenced by the 3D surface plots. The highest AgNPs yield was achieved at a high temperature of 30–35 °C.

The 3D surface graphs (Figure 4C,E,F) illustrate the effect of incubation period on AgNPs biosynthesis using the aqueous mycelial-free filtrate of *A. flavus* when interacting with the other three factors: AgNO_3_ concentration, initial pH level and temperature; respectively. The 3D surface plots indicate a positive correlation between the incubation time and AgNPs biosynthesis. AgNPs biosynthesis increased by increasing the incubation time. The highest AgNPs yield was obtained toward the high level of the incubation time (56–71 h).

#### 3.4.2. The Model’s Adequacy

In Figure 5A, the normal probability plot (NPP) illustrates that the data are normally distributed since the residuals are in line with a straight line. This indicates that the experimental results are consistent with the predicted data, which confirms the model’s accuracy [67].

Figure 5B shows the actual against predicted values of AgNPs biosynthesis. Figure 5B illustrates that all data points are closely aligned with the prediction line. This signifies an appropriate correlation between the experimental and predicted data [68].

The Box-Cox plot in Figure 5C shows the model transformation for the biosynthesis of AgNPs. The best lambda (*λ* = 0.99) is marked by the green line in Figure 5C, whereas the blue line indicates the current lambda (*λ* = 1). The lowest and highest values of the 95% confidence interval, shown by red lines, ranges from 0.91 to 1.07. The current *λ* locates between the two vertical red lines which suggests that the model fits well the experimental data and no need for data transformation [68].

Figure 5D represents the residual against the predicted values of AgNPs biosynthesis. Figure 5D demonstrates the goodness-of-fit of the model, displaying randomly scattered residuals around the zero-line with a uniform distribution. This suggests that the experimental results variance is constant for all values, which are crucial for statistical precision [69].

#### 3.4.3. The Desirability Function

The optimal conditions for maximal AgNPs biosynthesis were theoretically predicted using the desirability function, which was experimentally validated. The desirability function value ranges from 0 (undesirable) to 1 (desirable) [70]. Desirability functions are typically calculated mathematically before experimental validation [71]. The optimum predicted conditions for maximizing AgNPs biosynthesis, using aqueous mycelial-free filtrate of *A. flavus*, has been determined through the desirability function as follows: incubation time (58.12 h), initial pH (7.99), AgNO_3_ concentration (4.84 mM/mL), and temperature (34.84 °C). The highest predicted value for the biosynthesized AgNPs was 235.72 µg/mL, as shown in Figure 6. The highest experimental value of AgNPs biosynthesis under these conditions was 247.53 µg/mL. The validation demonstrated the high accuracy of the applied model.

### 3.5. Characterization of the Biosynthesized AgNPs

The present study characterized AgNPs synthesized by *A. flavus* using diverse methods, including UV-visible spectroscopy, scanning electron microscopy (SEM), transmission electron microscopy (TEM), selected area diffraction pattern (SADP), energy-dispersive X-ray spectroscopy (EDX), zeta potential analysis, X-ray diffractometer (XRD) and FTIR transform infrared attributes.

The application of SEM in the investigation of AgNPs provided insights into the size and surface features of the NPs. The results proved that the AgNPs had uniformly sized spherical particles as observed in Figure 7A, as opposed to any irregular shape, and were free of agglomeration. The sizes of the synthesized nanoparticles varied between 1–50 nm, although a few particles displayed signs of clustering [72].

The TEM micrographs revealed the morphology and size distribution of the particles, ranging from 8.93 to 19.11 nm. The micrograph depicts both individual and aggregated particles, with a capping agent maintaining separation and causing repulsion between protein-based NPs. Spherical nanosilver particles were observed (Figure 7B). Particles of varied shapes and sizes, similar to biological systems, are commonly observed [73]. Fouda, et al. [13] reported synthesized Ag-NPs ranging from 3 to 28 nm, averaging 12.5 ± 5.1 nm. Raza, et al. [74] found that 15 nm spherical nanoparticles were more toxic to Gram-negative bacteria *Escherichia coli* and *Pseudomonas aeruginosa* than larger spherical and triangular shapes. Therefore, we expect the AgNPs synthesized in this study to exhibit high activity owing to their smaller size.

Crystallographic experiments, including selected area diffraction pattern (SADP) analyses, were performed TEM. The SADP technique enables the investigation of separate droplets from nano-colloidal mixtures, revealing the diffraction pattern of nano-silver as luminous points against a dark field. The presence of distinctive diffraction rings in this pattern verifies the crystalline nature and successful production of nano silver [64], as illustrated in Figure 7C.

Energy-dispersive X-ray spectroscopy (EDX) was employed to analyze the characteristics, distribution, and concentration of elements in AgNPs using TEM as illustrated in Figure 7D. EDX spectrum provided strong evidence for silver as the dominant element in the AgNPs. The fabrication process of AgNPs may cause structural changes; therefore, EDX can help detect variations. EDX spectrum showed peaks around 3.40 keV, corresponding to AgL [75]. As shown in Figure 7D, these peaks might also be attributed to biomolecules adhering to the AgNPs surface.

To assess the repulsive forces between the generated AgNPs, the zeta potential of the biogenic AgNPs was measured. As shown in Figure 8A, a single peak with a value of +1.69 mV indicated repulsion among the particles. The zeta potential is of paramount significance for ensuring the stability of nanoparticles. Highly positive or negative zeta potential values result in strong repulsion, which effectively prevents aggregation [76]. Conversely, low zeta potential values indicate weak repulsive forces, facilitating particle aggregation and cluster formation.

X-ray diffraction (XRD) analysis is considered one of the simplest, universally recognized analytical techniques, to prove the construction of nanoparticles, and evaluate its crystal structures [77]. As presented in Figure 8B, the XRD diffractogram of AgNPs, showed brilliant crystal quality and revealed the presence of 4 discrete intense diffraction peaks at 2θ° over the entire spectrum, which ranged from 20 to 80 nm, hence validating the achievement of great purity. The peaks were detected at 27.5, 32, 45.9, and 54.6 which closely coincide to crystal planes 210,113,124 and 142; respectively [78,79]. In consistent with our result Majeed, et al. [80] found that the XRD analysis pattern of AgNPs presented sharp diffraction 2θ° at 27, 32, 46, 54, 57, and 67. Consequently, the highly strong peak of the XRD pattern has confirmed that the resultant particles in the sample are in the form of crystalline silver nanoparticles with face centered cubic structure [81]. Hence, the XRD analysis has revealed that the aqueous mycelial-free filtrate of *A. flavus* has the ability to reduce silver ions and synthesized AgNPs with well-defined dimensions.

FTIR spectroscopy identified functional groups on AgNPs that are essential for stabilizing and reducing silver ions. FTIR analysis of the synthesized AgNPs showed characteristic vibrational frequencies at 3448, 2960, 2929, 1801, 1633, 1025, and 522 cm^−1^, as depicted in Figure 8C. These peaks likely correspond to the functional groups involved in the biomolecular interactions, stabilization, and reduction of AgNPs. The peaks at 3448 and 2960 cm^−1^ correspond to the N–H stretching of primary and secondary amides of proteins and O–H stretching of alcohols and phenols, respectively. The 2929 cm^−1^ peak corresponds to C–H symmetrical stretching of alkanes. The 1801 cm^−1^ peak corresponds to the C=O in carboxylic acids. The peak at 1633 cm^−1^ corresponds to carbonyl stretching of the amide I bond in proteins. The peaks at 1025 cm^−1^ and 522 cm^−1^ correspond to aliphatic amines and C–Cl stretching of alkyl halides, respectively. These findings indicate the presence of various functional groups in the aqueous mycelial-free filtrate of *Aspergillus flavus*, such as alkanes, alkenes, aliphatic and aromatic amines, and alkyl groups, which play significant roles in the reduction, capping, and stabilization of AgNPs. Previous studies on AgNPs synthesized by *Trichoderma longibrachiatum* identified unique FTIR peaks at 1634.92, 2156.94, and 3269.31 cm^−1^, which are linked to primary amines [82] and protein structures [83]. The band around 1651 cm^−1^ signifies protein amide bonds (bending vibrations), whereas the 3448 cm^−1^ peak indicates primary amine stretching vibrations [84]. These FTIR findings suggest that phenols and proteins act as reducing, stabilizing, and capping agents for AgNPs, facilitating the transformation of silver radicals into silver ions [85].

### 3.6. Antimicrobial Efficacy of Cotton Fabrics Loaded with Bio-Synthesized AgNPs

The agar well diffusion method assessed the antimicrobial efficacy of AgNPs-embedded cotton fabric against *E. coli*, *S. aureus*, and *C. albicans* (Figure 9). A 24 h incubation revealed clear inhibition zones, demonstrating the release of the antimicrobial agent. The AgNPs-treated fabric exhibited inhibition zones of 15, 12, and 17 mm against *E. coli*, *S. aureus*, and *C. albicans*, respectively, indicating AgNPs diffusion into the agar. The inhibitory mechanism of Ag^+^ ions involves interactions with negatively charged cell membranes through electrostatic forces [86,87]. Patil, et al. [28] found that AgNPs-coated cotton fabrics showed significant antibacterial activity against Gram-positive and Gram-negative bacteria, with larger inhibition zones confirming efficacy. Ibrahim and Hassan [88] reported strong antibacterial activity for AgNPs -treated fabrics at all doses, with inhibition zones of 13–17 mm, even at the lowest AgNPs concentration (1 mM). Antibacterial activity was more pronounced against *E. coli* than against *S. aureus*. The AgNPs -incorporated cotton fibers produced in this study consistently exhibited inhibition zones greater than 12 mm.

AgNPs possess advantages such as diminutive dimensions and increased surface area, facilitating bacterial cell penetration and nuclear access [89,90]. Studies have highlighted the role of electrostatic attraction between negatively charged bacterial cells and positively charged AgNPs in antibacterial effectiveness [43,71]. Our findings indicated the greatest inhibition zone against *C. albicans*, followed by *E. coli* and *S. aureus*, consistent with Fatima, et al. [91], who reported that AgNPs synthesized using a specific *A. flavus* strain demonstrate antibacterial and antifungal activity against *Trichoderma* sp.

### 3.7. Molecular Docking Analysis of AgNPs Biosynthesized by the Aqueous Mycelial-Free Filtrate of A. flavus Loaded onto Cotton Fabrics

Both rule-based and machine-learning approaches have been employed in forward synthesis tools. These methods are used to identify the reactions from the available in silico list that can generate product structures. As depicted in Figure 10, the forward interaction between cotton fabric cellulose and the biosynthesized AgNPs was demonstrated. From the results, the cellulase is predicted to break down by AgNPs and form smaller polymer units that can spread into microbial environments. The resulting polymers can attach to proteins in microbes, thereby suppressing microbial proliferation. Combining innovative synthetic approaches with computer-aided retrosynthesis and forward synthesis methods can significantly expand the structural variety of compounds that are feasible for synthesis. These approaches can encompass a large portion of the relevant chemical reactions by utilizing extensive unbiased reaction databases [92]. In this investigation, in silico analysis of predicted monomer compound derived from biosynthesized AgNPs loaded on cotton fabric exhibited the potential to inhibit bacterial and *Candida* proliferation, the binding energies between predicted monomer compound and various receptor proteins linked to antibiotic and antifungal resistance in bacteria and fungi including penicillin-binding proteins (PBPs) and Lanosterol-14-demethylase (L-14α-DM), showed in Table 4.

The study showed that integrating biosynthesized silver nanoparticles with cotton fabric resulted in antimicrobial effects against both Gram-positive and Gram-negative bacteria, including *Staphylococcus aureus* and *E. coli*, respectively. These outcomes are consistent with the work of Hashem, et al. [93], which indicated that cellulose modified with nanoformulations successfully impeded the growth of gram-negative bacteria (*Pseudomonas aeruginosa* and *E. coli*) and Gram-positive bacteria (*Staphylococcus aureus* and *Bacillus subtilis*). Altering the structure of cellulose provides practical approaches to changing the characteristics of this material, facilitating its application in various fields while preserving its environmentally friendly and renewable nature. In the near future, the adoption of these technological innovations could lead to the development of novel and eco-friendly materials [22].

The results also demonstrated that the predicted compound derived from synthesized AgNPs attached to cotton fabrics exhibited stronger binding energy with L-14α-DM proteins than PBPs. The marginally higher binding energy of L-14α-DM (−5.2 kcal/mol) than PBPs (−4.7 kcal/mol) indicates a more robust interaction with this protein, which is attributed to the presence of a covalent hydrogen bond with an unfavorable donor bond involving amino acid residues (Figure 11 and Table 4).

Penicillin-binding proteins (PBPs) are a group of molecules that exhibit strong attraction to the β-lactam antibiotic, penicillin. These membrane-bound proteins are vital for the synthesis of peptidoglycan, which is the primary structural component of bacterial cell walls. The significance of PBPs lies in their diverse roles, including protein-protein interactions, antibiotic resistance mechanisms, cell wall construction, and various regulatory functions [94]. β-lactams specifically attach to the active site through covalent bonding or to both the allosteric site via non-covalent bonding and the active site through covalent bonding [95]. Current docking studies suggest that the moderate binding affinity between the compound predicted from AgNPs and cotton fabric could potentially disrupt PBPs function and hinder peptidoglycan cross-linking in the bacterial cell wall, resulting in cellular death. As noted in Farid, et al. [96], the interaction with arginine and glutamic acid residues indicated the existence of electrostatic or hydrogen bond interactions between the predicted compound and the PBPs protein, while the interaction with proline may help stabilize the connection between proteins and the predicted compound derived from AgNPs with cotton fabric. However, developing an effective antifungal therapeutic agent is more challenging because of the eukaryotic nature of *Candida*. A range of pharmacological compounds is currently used to treat *Candida* infections, each targeting different essential factors [97].

This study revealed that the compound predicted from AgNPs -loaded cotton fabrics exhibited binding affinity with lanosterol-14-demethylase (L-14α-DM) through molecular docking interactions. L-14α-DM serves as an excellent therapeutic target for a wide range of anticandidal and antifungal treatments [98]. These findings indicate a more robust interaction with PBPs, suggesting substantial interference with lanosterol-14-demethylase (L-14α-DM) and disruption of ergosterol production in the cell membrane, ultimately causing cell death, which is in line with a previous report. Many studies have shown that one factor contributing to *C. albicans* antibiotic resistance is a mutation in the L-14α-DM enzyme [99].

Our molecular docking analysis verified that the interruption of sterol biosynthesis leads to cell death. The interaction between serine and methionine may help stabilize the predicted compound near the active site through hydrogen bond formation, whereas the interaction between methionine and its sulfur side chain aids the predicted compound from AgNPs with cotton fabric in inhibiting enzyme and histidine function. This observation aligns with that of Ganeshkumar, et al. [100], who noted that *C. albicans* amino acid residues, specifically SER 378, SER 378, HIS 377, and MET 508, exhibited strong interactions with posaconazole and VT-1161, resulting in cell death. Furthermore, the systematic development of environmental substances that lack skin-sensitizing properties is in line with the green chemistry principles [101]. Computational approaches offer an appealing method for creating safer chemical compounds [102].

### 3.8. Pharmacokinetic Properties of the Predicted Compound Resulting from AgNPs Loaded onto Cotton Fabrics

Computational methods were applied to explore the pharmacological attributes and toxicity of the predicted compound derived from cotton fabrics loaded with AgNPs (Appendix A). The findings revealed that the compound had poor absorption in the gastrointestinal tract, suggesting limited effectiveness for oral administration, as noted by Martins, et al. [103]. Furthermore, the compound’s log Kp value for skin permeability was determined to be −9.49, indicating minimal penetration through the skin, which aligns with the results reported in [104]. The compound’s pharmacokinetic profile suggested it could be a promising candidate for additional research. The KeratinoSens assay results classified the test compound as a non-sensitizer, suggesting it does not trigger skin sensitization by interfering with the Keap1-Nrf2-ARE signaling pathway in differentiated human keratinocytes (Figure 12A). To enhance the use of existing data, this study combined previous HRIPT and HMT information with IRR and MRL when examining allergen responses. Based on predictions from the Human Repeat Insult Patch Test (HRIPT) or Human Maximization Test (HMT), the compound is expected to be a non-sensitizer. This indicates that it is not likely to induce skin sensitization in humans when tested under the conditions specified in these assessments (Figure 12B).

The compound currently predicted from AgNPs deposited on cotton fabric in the computational KeratinoSens assay demonstrates chemical and structural characteristics that are consistent with the model’s prediction range. This alignment indicates a strong likelihood of obtaining reliable results. A confidence level of 94% indicates a high probability that the substance will not induce skin sensitization. An accurate probability map of non-sensitizers is unlikely to contain groups comprising reactive centers or chemotypes capable of initiating stress signals in keratinocytes. The molecule does not exhibit electrophilic positions or other functional groups that could potentially participate in the formation of covalent bonds with skin proteins during the sensitization process [105].

When predicting the results of HRIPT and HMT for substances derived from cotton fabrics containing AgNPs, the Applicability Domain, and Confiability suggest that the substance’s structure or characteristics fall outside the range the predictive model was trained on. Therefore, caution is warranted when interpreting the result, as the model may lack sufficient data to accurately predict the effect of this compound. A reliability of 94.9% indicates a high level of confidence in the result, although the compound was found to be outside the AD, which aligns with the conclusions drawn by Herzler et al. [106].

## 4. Conclusions

The research findings demonstrated an easy, affordable, and environmentally friendly method for the synthesis of AgNPs using the extracellular filtrate of the *Aspergillus flavus* fungal strain and an aqueous solution of silver nitrate. This method is environmentally friendly because it does not require toxic chemicals or organic solvents. Our results demonstrated that the synthesized AgNPs exhibited promising antimicrobial properties against specific bacteria and fungi, suggesting their potential applications in biomedicine. These findings indicated that they are particularly important from a technological aspect for producing antibacterial finishes. This treatment is suitable for the manufacture of antimicrobial coatings and fabrics. More research should be conducted on curing processes, which play a significant role in the stabilization of AgNPs applied on fabric surfaces. In addition to the in silico approach, a compound predicted from AgNPs deposited on cotton fabric showed promising pharmacokinetic properties, particularly as a non-skin sensitizer with minimal potential for drug interactions.

## Figures and Tables

**Figure 1 jfb-15-00354-f001:**
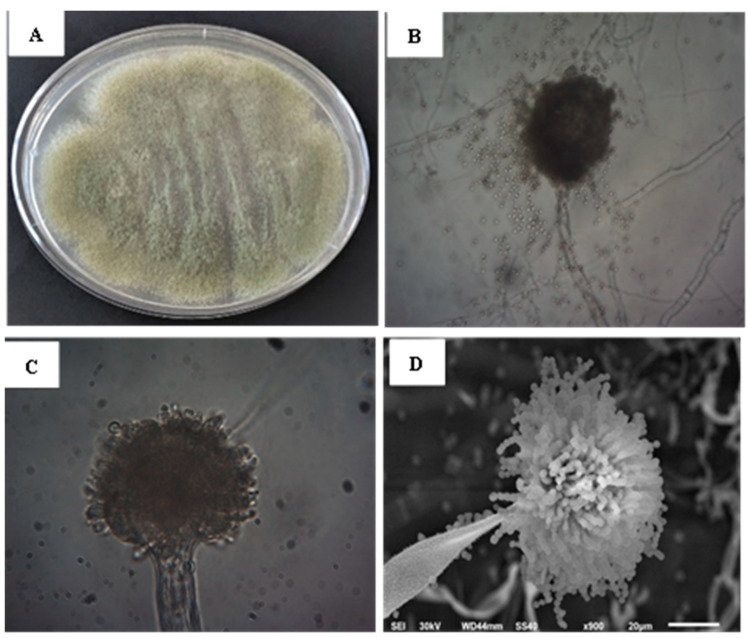
Identification of *Aspergillus* via morphological and structural analysis: (**A**) Characteristic growth on PDA medium after 7 days at 25 °C; (**B**,**C**) Microscopic views at 100× and 400× magnification, displaying septate branched mycelium with conidia; (**D**) SEM imaging.

**Figure 2 jfb-15-00354-f002:**
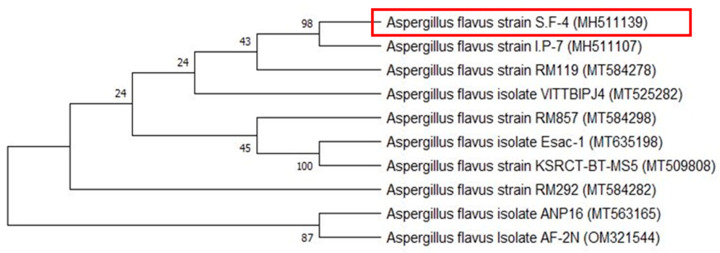
A construct of the phylogenetic tree of *Aspergillus* sp. based on internal transcribed spacer (ITS) region sequences with 1000 bootstrap replicates. The accession numbers are indicated in parentheses and the red box indicates the studied strain.

**Figure 3 jfb-15-00354-f003:**
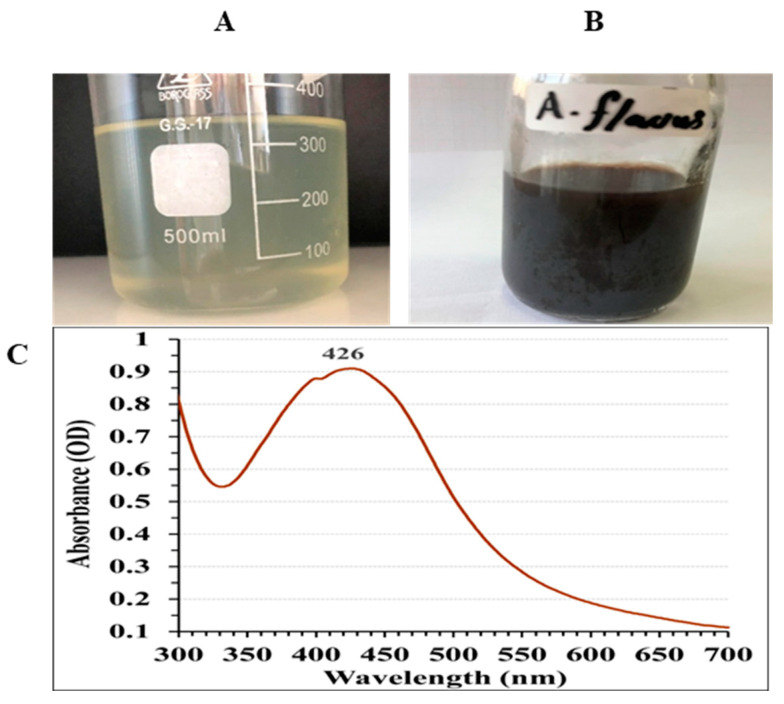
Production of AgNPs using the aqueous mycelial-free filtrate of *A. flavus*. (**A**) Control flask (the aqueous mycelial-free filtrate without silver ions), (**B**) Experimental flask (the aqueous mycelial-free filtrate with silver ions) following 72 h cultivation, (**C**) Ultraviolet-visible absorption spectrum of the synthesized AgNPs (300–700 nm).

**Figure 4 jfb-15-00354-f004:**
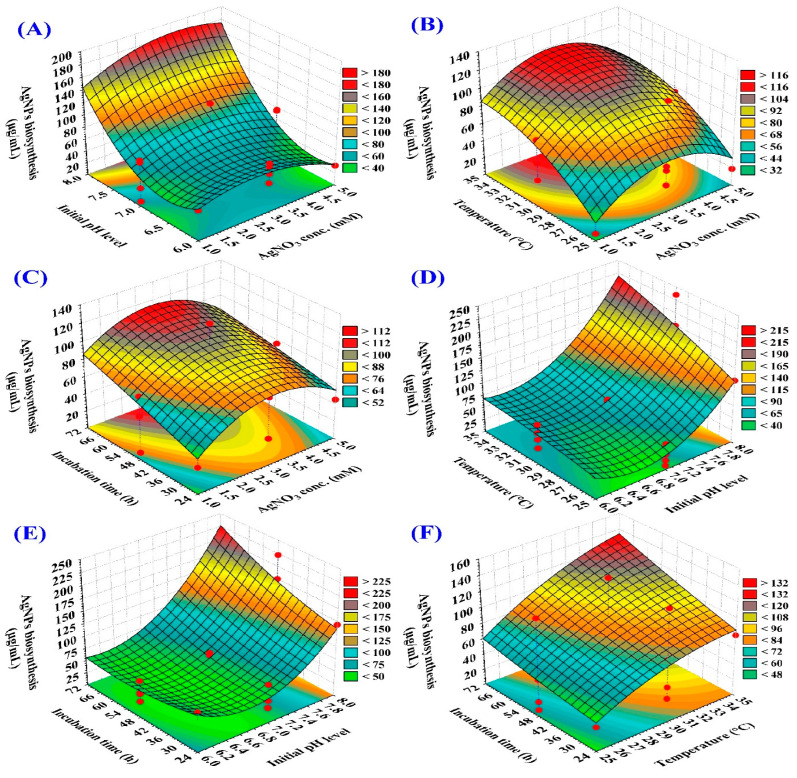
3D plots illustrating the interactive impacts of AgNO_3_ concentration (X_1_), starting pH value (X_2_), temperature (X_3_), and incubation time (X_4_) on the biosynthesis of AgNPs using the aqueous mycelial-free filtrate of *A. flavus*. (**A**–**C**) illustrate the effect of AgNO_3_ concentration on AgNPs biosynthesis when interacting with initial pH level, temperature and incubation period; respectively. (**A**,**D**,**E**) illustrate the effect of initial pH level on AgNPs biosynthesis when interacting with AgNO_3_ concentration, temperature, and incubation time; respectively. (**B**,**D**,**F**) illustrate the effect of temperature on AgNPs biosynthesis when interacting with the AgNO_3_ concentration, initial pH level and incubation time; respectively.

**Figure 5 jfb-15-00354-f005:**
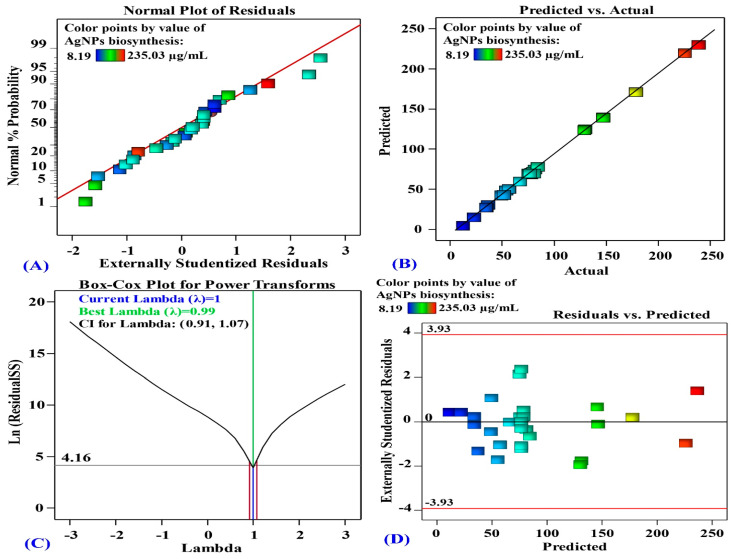
(**A**) Normal probability plot of internally studentized residuals, (**B**) plot of predicted versus actual, (**C**) Box-Cox plot of model transformation and (**D**) plot of internally studentized residuals versus predicted values of AgNPs biosynthesis using aqueous mycelial-free filtrate of *A. flavus* as affected by AgNO_3_ conc. (X_1_), initial pH level (X_2_), temperature (X_3_) and incubation time (X_4_).

**Figure 6 jfb-15-00354-f006:**
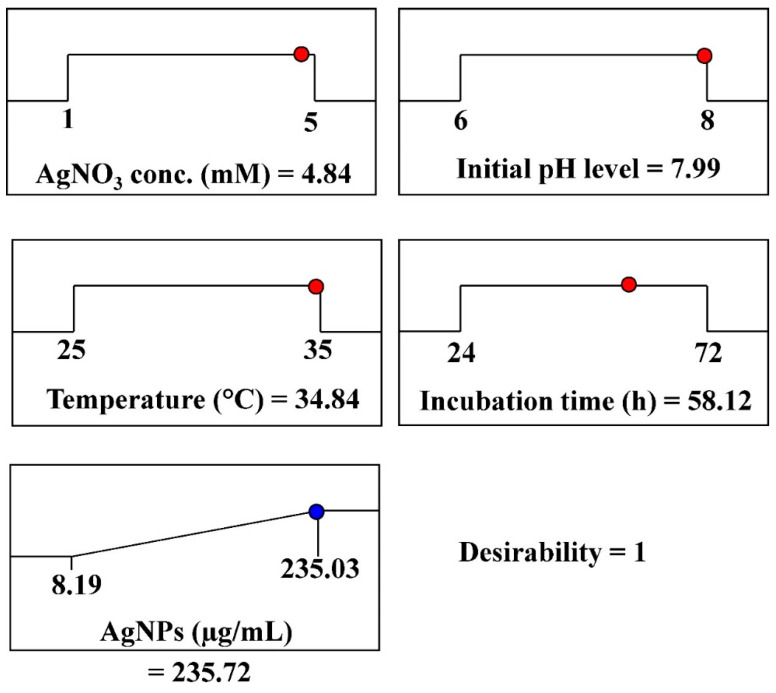
The optimization plot displays the desirability function and the optimal predicted values for the synthesis of AgNPs using aqueous mycelial-free filtrate of *A. flavus*. The red and blue circles represent the highest values for the variables and AgNPs; respectively.

**Figure 7 jfb-15-00354-f007:**
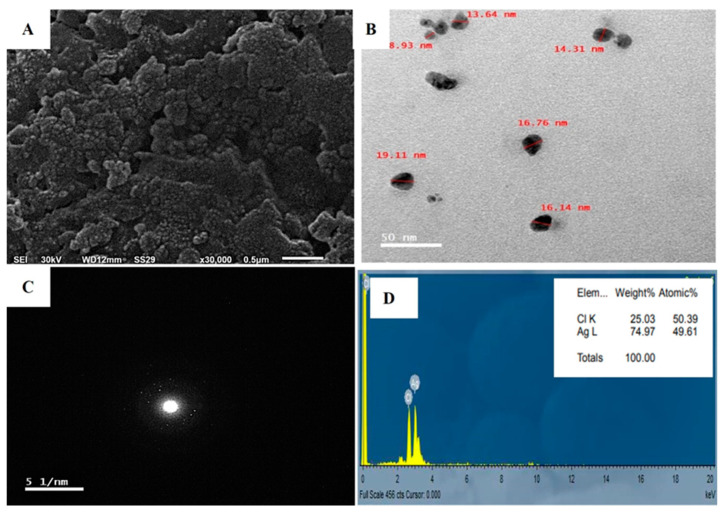
Biogenic AgNPs by *A. flavus* comprising: (**A**) SEM image, (**B**) TEM micrograph, (**C**) SADP for a single nanosilver particle, and (**D**) EDX examination illustrating the elemental composition of native silver.

**Figure 8 jfb-15-00354-f008:**
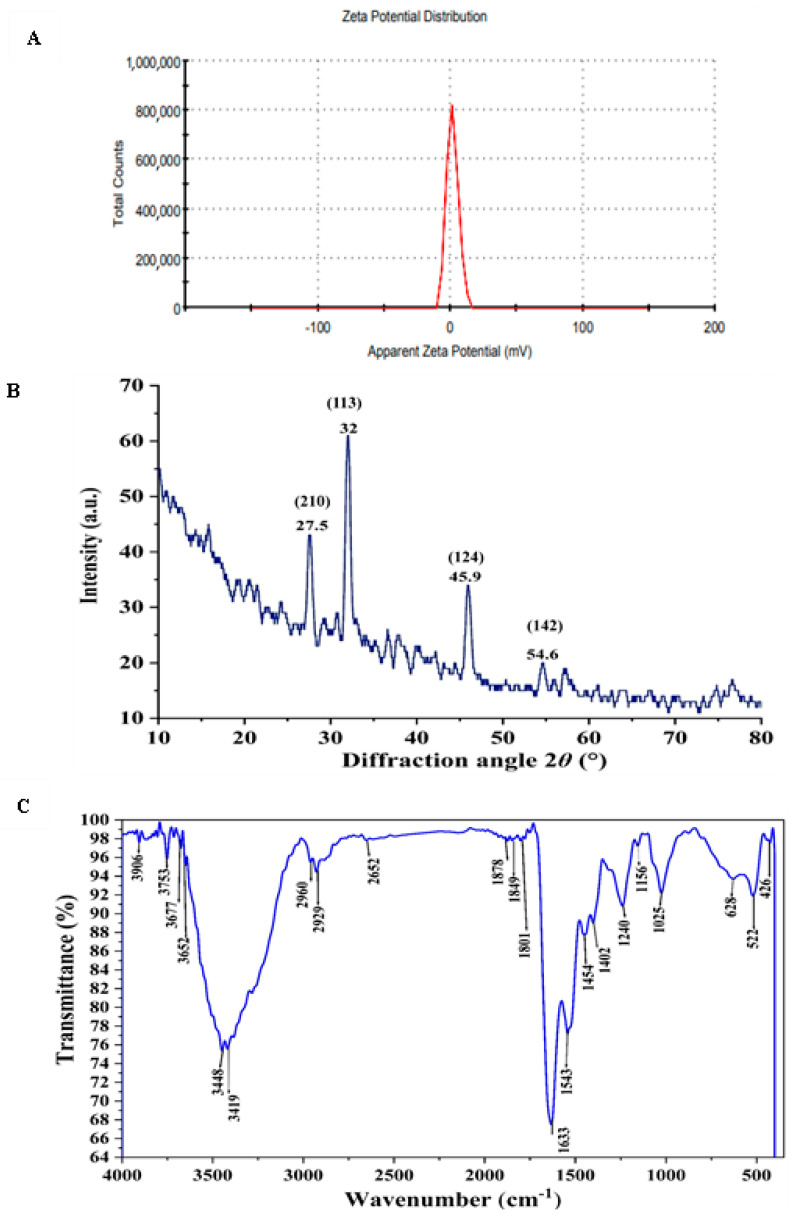
Analysis of biogenic AgNPs using (**A**) Zeta potential measurement, (**B**) XRD pattern of silver nanoparticles and (**C**) FTIR spectroscopy to identify functional groups that stabilize or cap AgNPs.

**Figure 9 jfb-15-00354-f009:**
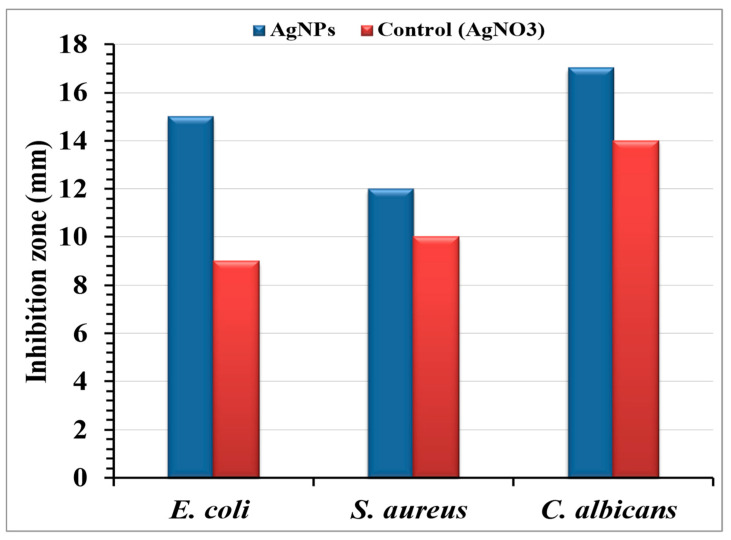
Antimicrobial activity of AgNPs bio-synthesized by the aqueous mycelial-free filtrate of *Aspergillus flavus* loaded on cotton fabrics.

**Figure 10 jfb-15-00354-f010:**
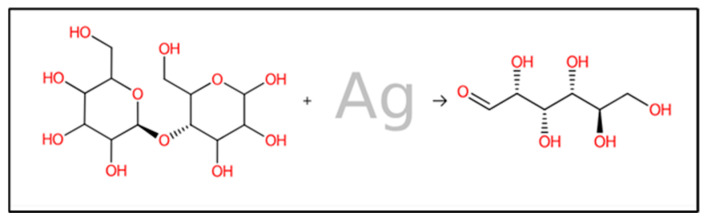
Prediction of forward reaction mechanism between bio-synthesized AgNPs and cellulose in cotton fabric.

**Figure 11 jfb-15-00354-f011:**
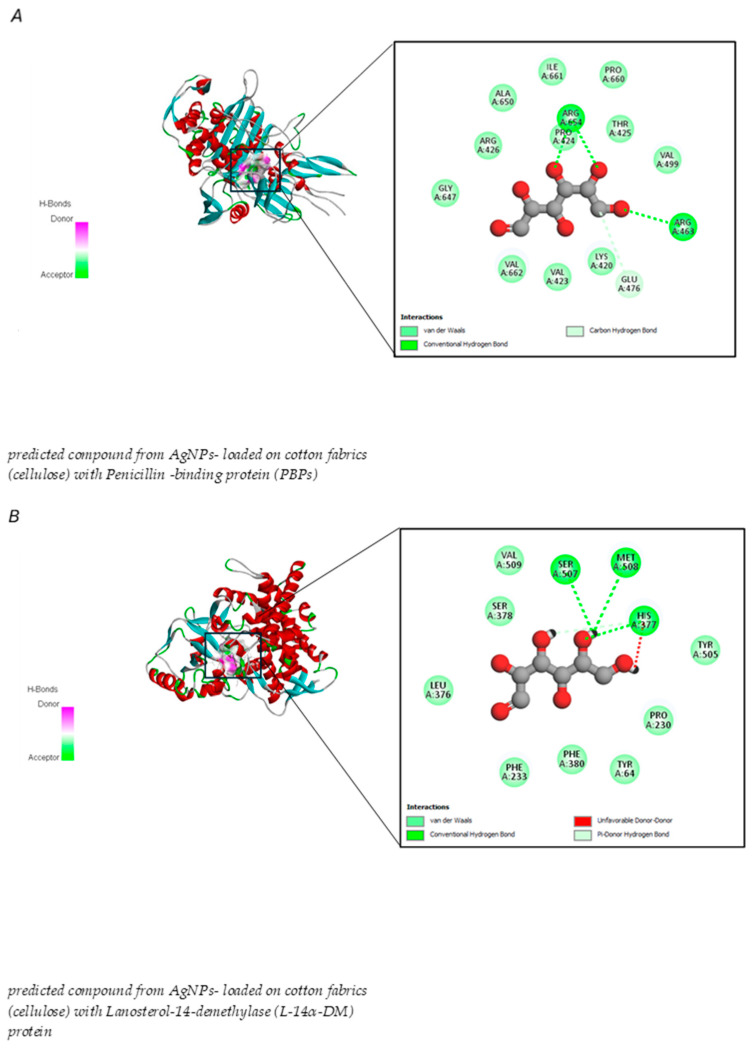
Molecular docking interactions between predicted compounds from AgNPs-loaded cotton fabrics (cellulose) with microbial proteins: (**A**) PBPs in Gram +ve and −ve bacteria, (**B**) Lanosterol-14α-demethylase (L-14α-DM) protein in *Candida albicans*.

**Figure 12 jfb-15-00354-f012:**
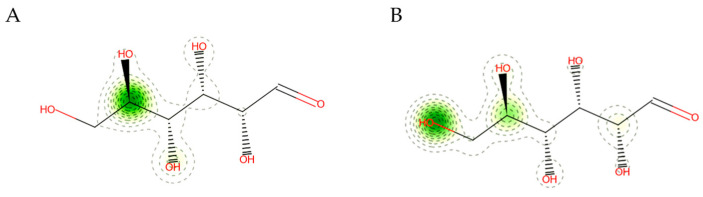
Sensitizer prediction of the predicted compound resulting from AgNPs loaded on cotton fabrics. (**A**) Prediction of keratinocyte responses to the predicted compound resulting from AgNPs loaded on cotton fabrics. (**B**) Prediction of human repeated insult patch test (HRIPT) and human maximization test (HMT) of the predicted compound resulting from AgNPs loaded on cotton fabrics.

**Table 1 jfb-15-00354-t001:** The Box-Behnken design matrix illustrates *A. flavus*-mediated biosynthesis of AgNPs, as affected by AgNO_3_ concentration (X_1_), initial pH (X_2_), temperature (X_3_), and incubation time (X_4_) with both coded and actual factor levels.

Std	Run	Variables				AgNPs (µg/mL)		Residuals
		X_1_	X_2_	X_3_	X_4_	Experimental	Predicted	
11	1	−1	0	0	1	76.0	75.23	0.77
14	2	0	1	−1	0	124.77	126.82	−2.05
6	3	0	0	1	−1	77.48	77.78	−0.30
27	4	0	0	0	0	71.26	72.88	−1.62
13	5	0	−1	−1	0	47.56	46.15	1.41
20	6	1	0	1	0	75.48	75.06	0.42
1	7	−1	−1	0	0	49.75	51.59	−1.84
4	8	1	1	0	0	174.72	174.31	0.40
12	9	1	0	0	1	74.1	73.7	0.4
16	10	0	1	1	0	221.69	222.75	−1.06
22	11	0	1	0	−1	143.2	142.24	0.96
18	12	1	0	−1	0	18.88	18.22	0.66
3	13	−1	1	0	0	142.84	142.84	0.00
9	14	−1	0	0	−1	30.32	30.34	−0.02
23	15	0	−1	0	1	62.99	62.87	0.12
26	16	0	0	0	0	71.1	72.88	−1.78
2	17	1	−1	0	0	32.33	33.77	−1.44
19	18	−1	0	1	0	72.51	72.09	0.42
21	19	0	−1	0	−1	80.19	80.86	−0.67
7	20	0	0	−1	1	52.47	53.61	−1.14
5	21	0	0	−1	−1	30.96	30.51	0.45
24	22	0	1	0	1	235.03	233.28	1.75
15	23	0	−1	1	0	74.03	71.62	2.41
8	24	0	0	1	1	125.85	127.74	−1.89
17	25	−1	0	−1	0	8.19	7.53	0.66
25	26	0	0	0	0	73.11	72.88	0.23
10	27	1	0	0	−1	45.11	45.53	−0.43
29	28	0	0	0	0	72.55	72.88	−0.33
28	29	0	0	0	0	76.43	72.88	3.55
11	1	−1	0	0	1	76.01	75.23	0.78
**Variable**		**Code**		**Coded and Actual Levels**				
				**−1**	**0**	**1**		
AgNO_3_ conc. (mM)		X_1_		1	3	1		
Initial pH level		X_2_		6	7	5		
Temperature (°C)		X_3_		25	30	8		
Incubation time (h)		X_4_		24	48	35		

**Table 2 jfb-15-00354-t002:** ANOVA of Box–Behnken experimental design used for synthesizing AgNPs using aqueous mycelial-free filtrate of *A. flavus* to evaluate the effects of four factors: AgNO_3_ concentration (X_1_), initial pH level (X_2_), temperature (X_3_), and incubation time (X_4_).

Source of Variance		Coefficient Estimate	Sum of Squares	Degrees of Freedom	Mean Square	*f*-Value	*p*-Value
Model	Intercept	72.88	85,288.54	14	6092.04	1734.16	<0.0001 *
Linear effect	X_1_	3.41	139.79	1	139.79	39.79	<0.0001 *
	X_2_	57.95	40,297.60	1	40,297.60	11,471.09	<0.0001 *
	X_3_	30.35	11,053.54	1	11,053.54	3146.50	<0.0001 *
	X_4_	18.26	4002.78	1	4002.78	1139.43	<0.0001 *
Interaction effect	X_1_ X_2_	12.32	607.53	1	607.53	172.94	<0.0001 *
	X_1_ X_3_	−1.93	14.90	1	14.90	4.24	0.0586
	X_1_ X_4_	−4.18	69.97	1	69.97	19.92	0.0005 *
	X_2_ X_3_	17.61	1240.68	1	1240.68	353.17	<0.0001 *
	X_2_ X_4_	27.26	2971.88	1	2971.88	845.97	<0.0001 *
	X_3_ X_4_	6.72	180.41	1	180.41	51.36	<0.0001 *
Quadratic effect	X_1_^2^	−22.93	3411.56	1	3411.56	971.13	<0.0001 *
	X_2_^2^	50.68	16,658.37	1	16,658.37	4741.96	<0.0001 *
	X_3_^2^	−6.73	293.42	1	293.42	83.53	<0.0001 *
	X_4_^2^	6.25	253.46	1	253.46	72.15	<0.0001 *
Error effect	Lack of Fit		30.47	10	3.05	0.65	0.735
	Pure Error		18.71	4	4.68		
*R* ^2^	0.9994						
Adj *R*^2^	0.9988						
Pred *R*^2^	0.9976						
Adeq Precision	167.47						

* Significant values, *f*: Fishers’s function, *p*: Level of significance.

**Table 3 jfb-15-00354-t003:** The fit summary results of Box–Behnken experimental design used for synthesizing AgNPs using the aqueous mycelial-free filtrate of *A. flavus* as influenced by AgNO_3_ concentration (X_1_), initial pH level (X_2_), temperature (X_3_), and incubation time (X_4_).

Sequential Model Sum of Squares					
Source	Sum of Squares	*df*	Mean Square	*f*-Value	*p*-Value *p*rob >*f*
Linear vs. Mean	55,493.71	4	13,873.43	11.16	<0.0001 *
2FI vs. Linear	5085.37	6	847.56	0.6162	0.7148
Quadratic vs. 2FI	24,709.46	4	6177.36	1758.45	<0.0001 *
**Lack of Fit Tests**					
**Source**	**Sum of Squares**	** *df* **	**Mean Square**	***f*-Value**	***p*-Value** ***p*rob >*f***
Linear	29,825.3	20	1491.27	318.83	<0.0001 *
2FI	24,739.93	14	1767.14	377.81	<0.0001 *
Quadratic	30.47	10	3.05	0.65	0.735
**Model Summary Statistics**					
**Source**	**Standard Deviation**	** *R* ^2^ **	**Adjusted *R*^2^**	**Predicted *R*^2^**	**PRESS**
Linear	35.26	0.6503	0.592	0.4558	46,441.64
2FI	37.09	0.7099	0.5487	0.0541	80,717.46
Quadratic	1.87	0.9994	0.9988	0.9976	204.75

* Significant values, *df*: degree of freedom, PRESS: Prediction Error Sum Square, 2FI: two-factor interaction.

**Table 4 jfb-15-00354-t004:** Molecular Docking Analysis of predicted compounds resulting from biosynthesized AgNPs loaded on cotton fabrics as antimicrobial agents.

Predicted Compound Derived From	Target Protein	Binding Affinity Kcal/mol	Amino Acids Residues
AgNPs-loaded on cotton fabrics (cellulose)	Penicillin-binding proteins (PBPs)	−4.7	AGR: A654, AGR: A463, PRO: A424, GLUA:476
	Lanosterol-14-demethylase (L-14α-DM) protein	−5.2	SER: A507, MET: A508, HIS: A377

## Data Availability

The original contributions presented in the study are included in the article, further inquiries can be directed to the corresponding author.

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
