# Peer review of "Green Fabrication of Silver Nanoparticles, Statistical Process Optimization, Characterization, and Molecular Docking Analysis of Their Antimicrobial Activities onto Cotton Fabrics"

_jfb, 2024, doi:10.3390/jfb15120354_

Round 1
Reviewer 1 Report
Comments and Suggestions for Authors
1. I am concerned about whether Ag particles are active under light conditions or not in the absence of light.
2. What is the reactive functional groups during antimi- crobial activities on silver nanoparticles loaed cotton fabrics.
Author Response
We would like to extend sincere thanks to our reviewer for constructive comments on our work. Now we have tried our best to revise the manuscript according to the comments. These changes will not influence the content and framework of the paper and highlighted the revised text in yellow.
Reviewer comment 1: I am concerned about whether Ag particles are active under light conditions or not in the absence of light.
Authors Response: Thank you for pointing this out. Light is absorbed and scattered exceedingly well by silver nanoparticles. The strong interaction of light with nanoparticles causes collective oscillation of conduction electrons on the silver metal surface. Surface Plasmon Resonance is the name for oscillation. Localized Surface Plasmons have a femto second life span and are formed primarily through radiative and non-radiative processes. The release of radiation is part of the process of radioactive decay.
Reviewer comment 2. What is the reactive functional groups during antimicrobial activities on silver nanoparticles loaded cotton fabrics.
Authors response: Thanks for your comment. The reactive functional groups are Hydroxyl groups (-OH) in cotton fabrics These groups can:
- Bind to silver nanoparticles and aid in their attachment to the cotton surface, allowing for controlled silver ion release.
- Enhance the interaction between AgNPs and microbial cells on the cotton surface, as hydroxyl groups can form hydrogen bonds with microbial cell walls.
Thank you very much for your insightful remarks and comments that helped us enhance the manuscript.
Reviewer 2 Report
Comments and Suggestions for Authors
This study focuses on the biosynthesis and characterization of silver nanoparticles (AgNPs) using Aspergillus flavus fungus, highlighting their potential in multifunctional fabrics. The synthesis was confirmed by UV-Vis spectroscopy, SEM, TEM, SAED, Zeta Potential, EDX, and FTIR. The Box-Behnken design optimized the process, with ANOVA analysis showing significant linear coefficients for silver nitrate, initial pH, temperature, and incubation time. The results demonstrated that the synthesized AgNPs exhibited promising antimicrobial properties against specific bacteria and fungi, suggesting their potential applications in biomedicine. In my opinion, it is recommended that this paper be published in the Journal of Functional Biomaterials after addressing the following issues.
1. "X-ray diffraction (XRD) confirmed the crystalline nature" is mentioned in the abstract, but the XRD pattern and results analysis cannot be found in the text.
2. Figures 1 (B) and (C) suggest adding a scale.
3. Figure 3. (C): Ultra- 450 violet-visible absorption spectrum of the synthesized AgNPs (200-600 nm). But, the image here shows a wavelength range of 300-700 nm.
4. The concentration of AgNPs is described in many places in the paper. Please describe in detail the test method for the concentration of AgNPs.
5. Why is there no scale in Figure 7 (A)? In line 642, "The results proved that the AgNPs had uniformly sized spherical particles as observed in Figure 7(A), as opposed to any irregular shape, and were free of agglomeration." Please clearly mark on the photo that there are no aggregated spherical particles. In line 644, "The sizes of the synthesized nanoparticles varied between 1-50 nm," how did this result come about?
6. "Figure 7. C) SADP for a single nanosilver particle". How many nanometers is the scale in the picture?
7. According to the results in Figure 7.D), is it possible that the nanoparticle in this sample is the AgCl?
Author Response
Reviewer comment: This study focuses on the biosynthesis and characterization of silver nanoparticles (AgNPs) using Aspergillus flavus fungus, highlighting their potential in multifunctional fabrics. The synthesis was confirmed by UV-Vis spectroscopy, SEM, TEM, SAED, Zeta Potential, EDX, and FTIR. The Box-Behnken design optimized the process, with ANOVA analysis showing significant linear coefficients for silver nitrate, initial pH, temperature, and incubation time. The results demonstrated that the synthesized AgNPs exhibited promising antimicrobial properties against specific bacteria and fungi, suggesting their potential applications in biomedicine. In my opinion, it is recommended that this paper be published in the Journal of Functional Biomaterials after addressing the following issues.
Authors response: Thanks for your positive comments. We would like to extend sincere thanks to our reviewer for constructive comments on our work. Now we have tried our best to revise the manuscript according to the comments. These changes will not influence the content and framework of the paper and highlighted the revised text in green.
Reviewer comment: 1. "X-ray diffraction (XRD) confirmed the crystalline nature" is mentioned in the abstract, but the XRD pattern and results analysis cannot be found in the text.
Authors response: Thank you for pointing this out. Sorry! We missed it. We added all the required information of XRD in each section of the manuscript.
Reviewer comment: 2. Figures 1 (B) and (C) suggest adding a scale.
Authors response: Thanks for your comment. We mentioned the micrographs were photographed at 100 and 400x magnification.
Reviewer comment: 3. Figure 3. (C): Ultra- 450 violet-visible absorption spectrum of the synthesized AgNPs (200-600 nm). But, the image here shows a wavelength range of 300-700 nm.
Authors response: Thanks for your comment. We fixed it to 300-700 nm in all text.
Reviewer comment: 4. The concentration of AgNPs is described in many places in the paper. Please describe in detail the test method for the concentration of AgNPs.
Authors response: Agree. The determination of AgNPs concentration in the suspension using an experimental technique and all the requested information has been added in the revised version as follows:
The concentration of AgNPs can be quantitatively determined based on the linear dependence between AgNPs concentration and UV-Vis absorbance up to 200 ppm (Hung et al., 2018). To quantify AgNPs using the UV-Vis method, a calibration curve must be generated from the absorption intensity values of the UV-Vis spectrum determined from a known concentration solution. The variations in the UV-Vis spectra are indicative of the AgNPs' particle size and the solution's color. Different AgNPs concentration series (10–100 μg/mL) was obtained by diluting AgNPs stock solution (500 μg/mL) with deionized water. Those standards were measured together with samples to obtain the corresponding UV-Vis and the calibration curve for calculating sample concentrations.
Hung, N. D., Nam, V. N., ThiNhan, T., & Dung, T. T. N. (2018). Quantitative concentration determination of silver nanoparticles prepared by DC high voltage electrochemical method. Vietnam Journal of Chemistry, 56(5), 553-558.
Reviewer comment: 5. Why is there no scale in Figure 7 (A)?
Authors response: Thanks for your comment. Sorry the figures were not combined. The scale is found at the end of the figure.
Reviewer comment: In line 642, "The results proved that the AgNPs had uniformly sized spherical particles as observed in Figure 7(A), as opposed to any irregular shape, and were free of agglomeration." Please clearly mark on the photo that there are no aggregated spherical particles.
Authors response: Thanks for your comment. We used in SEM protocol AgNPs as a powder so it must appear as it is in Figure 7A, while in the TEM micrograph as shown in Figure 7B, there is no aggregation in the AgNPs solution.
In line 644, "The sizes of the synthesized nanoparticles varied between 1-50 nm," how did this result come about?
Authors response: Thanks for your comment. TEM micrographs revealed the morphology and size distribution of the particles, ranging from 8.93 to 19.11 nm.
Reviewer comment: 6. "Figure 7. C) SADP for a single nanosilver particle". How many nanometers is the scale in the picture?
Authors response: Thanks for your comment. The scale of SADP is 51 nm.
Reviewer comment: 7. According to the results in Figure 7.D), is it possible that the nanoparticle in this sample is the AgCl?
Authors response: Thanks for your comment. Yes, it is. In the green synthesis, AgCl is usually formed, it may be from fungal extract or sometimes from the used silver nitrate. Chloride ion (Cl−) is a common and abundant anion with a wide range of concentration in aquatic environments and exhibits a strong affinity for silver. AgNPs experienced multistep chlorination, which was dependent on the concentration of Cl− in a non-linear manner. The dissolution of AgNPs was accelerated at Cl/Ag ratio of 1 and the intensive etching effect of Cl− contributed to the significant morphology changes of AgNPs. The dissolved Ag+ quickly precipitated with Cl− to form an amorphous and passivating AgCl(s) layer on the surface of AgNPs, thus the dissolution rate of AgNPs decreased at higher Cl/Ag ratios (100 and 1000).
Thank you very much for your insightful remarks and comments that helped us enhance the manuscript.
Reviewer 3 Report
Comments and Suggestions for Authors
After reading the manuscript, I had several questions and suggestions. In addition, unfortunately, the text contains a number of inaccuracies and typos that need to be corrected. After that, the manuscript will be suitable for publication.

Author Response
Reviewer comment: First of all, I would like to express my respect for your hard work. I think the manuscript is generally well written and presents the research findings well, but unfortunately it needs the same clarifications and corrections that I outlined in my review.
Authors response: Thanks for your positive comments. We would like to extend sincere thanks to our reviewer for constructive comments on our work. Now we have tried our best to revise the manuscript according to the comments. These changes will not influence the content and framework of the paper and highlighted the revised text in violet.
Reviewer comment: 1. The material presents in paragraphs of the subsection “3.6. Molecular docking analysis of AgNPs biosynthesized by A. flavus loaded onto cotton fabrics” (lines 781-800) provides a rationale for why cellulose was chosen for the study. It does not fit into the Results and Discussion section and should be moved to the Introduction section.
Authors response: Thanks for catching this point. We moved it to the introduction section.
Reviewer comment: 2. The information in Figures 1A and D is sufficient to identify A. flavus. The material in Figures 1B and C is redundant and only supplements the main body of the manuscript.
Authors response: Thanks for your comment. We used in this study different methodology in identification of Aspergillus such as cultural, microscopic under light and electron microscopes and confirmed by molecular technique. Morphological identification of A. flavus is very important both under light and electron microscopes.
Reviewer comment: 3. It is unclear in what units the absorbance is presented in Figure 3.
Authors response: Thanks for your comment. The optical density (OD) is the unit and is found in the y axis on the chart.
Reviewer comment: 4. In Figure 7 it is not marked where A, B, C and D are. In addition, the quality of the EDX examination illustration is very low. This needs to be corrected. Additionally, it is necessary to place a table insert with the elemental composition of AgNPs in it.
Authors response: Thanks for your comment. We agreed with you. We modified it and became clearer and combined all figures A, B, C, and D together.
Reviewer comment: 5. The results of the tests of the antibacterial activity of nanoparticles, presented in Figure 9, are not informative. Instead of Figure 9 and Table 4, a figure with the data from Table 4 in column form should be provided. On the ordinate axis, mark the zone of antibacterial activity in mm, on the abscissa axis - microorganisms.
Authors response: Thanks for your comment. We agree with you. We performed the experiment again and mixed the table and figure data then draw a new chart.
Reviewer comment: 6. Please write in your manuscript what this means “AgL” (line 667).7. Unfortunately, the manuscript contains numerous typographical and proofreading errors that need to be corrected.
Authors response: Thank you for the notifications. We fixed all the typographical and proofreading problems in the paper. Our paper was revised by a native speaker of the language.
Reviewer comment: In the titles of Figures 1, 11 and Figures 3, 5, 8 12, their individual parts are written differently. For example.:
Figure 1. (Lines 393-395) A) Characteristic growth on PDA …; B and C) Microscopic views …; D) SEM…
Figure 3. (Lines 449-450) (A): Control flask… , (B): Experimental flask…, (C): Ultraviolet …
Authors response: Thanks for your comment. We revised the journal rules and fixed them all.
Reviewer comment: Subsection titles written in different styles. They should be written according to the journal's requirements. For example.:
“2.7.1. Ligand Preparation” and “2.5.1. Ultraviolet-Visible (UV-Vis) Spectral Analysis“ and so on.
Authors response: Thanks for your comment. We corrected it.
- Typos:
Reviewer comment: Line 197: “sterile dH2O”. It should be: “sterile deionized H2O”.
Authors response: Thanks for your comment. We added it.
Reviewer comment: Lines 203, 366, 419, 428 “Ag+ ions”. It should be: “Ag+”.
Authors response: Thanks for your comment. We corrected them.
Reviewer comment: Line 419 “Ag0”. It should be: “Ag0”.
Authors response: Thanks for your comment. We changed it.
Reviewer comment: Line 302 “formula C12H22O11, was retrieved…”. It should be: “C12H22O11”.
Authors response: Thanks for your comment. We modified it.
Reviewer comment: Line 554 “variables as shown in Figure 4. According…”. It should be: “according”
Authors response: Thanks for your comment. We rewrite this part.
Reviewer comment: Lines 580, 629 “ AgNO3 concentration...”. It should be: “AgNO3”.
Authors response: Thanks for your comment. We corrected it.
Reviewer comment: Line 634, “Figure 6. Figure 6. The optimization…”
Authors response: Thanks for your comment. We removed it.
Reviewer comment: Line 758, “Figure (10)”. It should be: “Figure 10”.
Authors response: Thanks for your comment. We modified it.
Reviewer comment: Please check the entire manuscript carefully.
Authors response: Thank you very much for your insightful remarks and comments that helped us enhance the manuscript.
Reviewer 4 Report
Comments and Suggestions for Authors
This paper deals with an easy, affordable and environmentally friendly method for the synthesis of AgNPs. Although interesting results were shown, it is insufficient at the present stage. Some suggestions are as follows.
Comment 1: Page 5, Line 217 “xj” should be also explained.
Comment 2: A great deal of data is presented by Tables and Figures. For simplicity, it is better to select the data that best conveys your findings and present them in Tables and Figures.
2-1: Figure 3 I don't understand the need for photos A and B.
2-2: Tables 1 and 2 Rather than presenting all of the data, it is better to consider about how to communicate to the reader what can be known from it.
2-3: Figure 6 Are these figures really necessary?
2-4: Figure 9 Can these results be quantified?
Author Response
Reviewer comment: This paper deals with an easy, affordable and environmentally friendly method for the synthesis of AgNPs. Although interesting results were shown, it is insufficient at the present stage. Some suggestions are as follows.
Authors response: Thank you for your kind considerations on our article! Now we have tried our best to revise the manuscript according to the comments. These changes will not influence the content and framework of the paper and highlighted the revised text in turquoise.
Reviewer comment: 1: Page 5, Line 217 “xj” should be also explained.
Authors response: Thanks for your comment. We added it in the abbreviations of the equation. : Levels of the independent variables
Reviewer comment: 2: A great deal of data is presented by Tables and Figures. For simplicity, it is better to select the data that best conveys your findings and present them in Tables and Figures.
Authors response: Thanks for your comment. All Tables and Figures must be represented and we added the extra data in the supplementary file.
Reviewer comment: 2-1: Figure 3 I don't understand the need for photos A and B.
Authors response: Thanks for your comment. As the green synthesis of silver nanoparticles is a stepwise process. Figure 3 A and B showed the colour changed from the A. flavus fungal extract to production of AgNPs.
Reviewer comment: 2-2: Tables 1 and 2 Rather than presenting all of the data, it is better to consider how to communicate to the reader what can be known from it.
Authors response: Thanks for your comment. These tables explained the Box-Behnken design matrix and its ANOVA of A. flavus-mediated biosynthesis of AgNPs, as affected by AgNO3 concentration (X1), initial pH (X2), temperature (X3), and incubation time (X4) with both coded and actual factor levels.
Reviewer comment: 2-3: Figure 6 Are these figures really necessary?
Authors response: Thanks for your comment. It is very important. Figure 6. defined the optimal predicted values for the synthesis of AgNPs using aqueous mycelial-free filtrate of A. flavus as a result of the optimization plot which displayed the desirability function.
Reviewer comment: 2-4: Figure 9 Can these results be quantified?
Authors response: Thanks for your comment. We quantified these results in Table 4 and explained in the results section.
Thank you very much for your insightful remarks and comments that helped us enhance the manuscript.
Round 2
Reviewer 3 Report
Comments and Suggestions for Authors
Thank you for your detailed answers. My comments have been taken into account and corrections have been made to the text of the manuscript. Overall, the manuscript is suitable for publication in the Journal of Functional Biomaterials.
Author Response
Thanks for your supportive comments. We appreciate the reviewer for agreeing with us on the importance and novelty of our study, as well as the important comments.
Reviewer 4 Report
Comments and Suggestions for Authors
This article can be published.
Author Response

(The authors gave the same response as above.)
